# An Assessment of ASTM E1922 for Measuring the Translaminar Fracture Toughness of Laminated Polymer Matrix Composite Materials

**DOI:** 10.3390/polym13183129

**Published:** 2021-09-16

**Authors:** Islam El-Sagheer, Amr A. Abd-Elhady, Hossam El-Din M. Sallam, Soheir A. R. Naga

**Affiliations:** 1Mechanical Design Department, Faculty of Engineering, Helwan University, Cairo 11718, Egypt; islam.ismaail@m-eng.helwan.edu.eg (I.E.-S.); amr_abdelfattah@m-eng.helwan.edu.eg (A.A.A.-E.); soheir_naga@m-eng.helwan.edu.eg (S.A.R.N.); 2Materials Engineering Department, Zagazig University, Zagazig 44519, Egypt

**Keywords:** ASTM E1922, ASTM D3039, laminated polymer matrix composite materials, translaminar fracture toughness, 3D FEM, Hashin criteria, Contour Integral Method

## Abstract

The main objective of this work is to predict the exact value of the fracture toughness (*K_Q_*) of fiber-reinforced polymer (FRP). The drawback of the American Society for Testing Materials (ASTM) E1922 specimen is the lack of intact fibers behind the crack-tip as in the real case, i.e., through-thickness cracked (TTC) specimen. The novelty of this research is to overcome this deficiency by suggesting unprecedented cracked specimens, i.e., matrix cracked (MC) specimens. This MC exists in the matrix (epoxy) without cutting the glass fibers behind the crack-tip in the unidirectional laminated composite. Two different cracked specimen geometries according to ASTM E1922 and ASTM D3039 were tested. 3-D FEA was adopted to predict the damage failure and geometry correction factor of cracked specimens. The results of the TTC ASTM E1922 specimen showed that the crack initiated perpendicular to the fiber direction up to 1 mm. Failure then occurred due to crack propagation parallel to the fiber direction, i.e., notch insensitivity. As expected, the *K_Q_* of the MC ASTM D3039 specimen is higher than that of the TTC ASTM D3039 specimen. The *K_Q_* of the MC specimen with two layers is about 1.3 times that of the MC specimen with one layer.

## 1. Introduction

Over the last decade, because of its appealing characteristics, such as high strength-to-weight ratio and excellent corrosion resistance, composite materials such as Glass Fiber Reinforced Polymer (GFRP) have noticed considerable growth in different applications. It is necessary to understand the fracture behavior of composite materials, such as matrix cracking, delamination, translaminar fracture, and fiber breakage, especially in aircraft structures. A fracture in a structure will tend to propagate and eventually lead to the structure collapse. This problem is resolved by fracture mechanics testing, which delivers information in critical stress intensity factors near the crack tip at fracture. Therefore, it needs to use accurate standard tests to measure the fracture properties of composite materials [1,2,3,4,5]. Compact tension specimen was used to determine the translaminar fracture toughness of carbon/epoxy composite laminates [2,3]. It was found that the major characteristics describing the fracture surfaces in the laminates comprised a mixture of the failure mechanisms, ply splitting, fibers bridging, and fiber pull-out. Haldar et al. [6] showed that the fiber bundle pull-out was the process that dissipates the most energy. Furthermore, Souza et al. [7] modified the ASTM E399 test method to account for the orthotropy of the composite materials and specimen geometry effects using a correction function based on a numerical evaluation of the strain energy release rate. They increased the initial notch depth from ≈16 mm to ≈23 mm to avoid compression and shear failures instead of mode I translaminar fracture under cyclic loading. Vantadori et al. [8] modified Jenq-Shah’s two-parameter model (MTPM) [9] to predict the fracture toughness of fiber-reinforced polymers (FRP) using three-point bend specimen according to RILEM [10]. It was found [8] that the predicted values of the fracture toughness and modulus of elasticity by MTPM are almost constant; consequently, such parameters are proved to be size-independent. The intrinsic and extrinsic mechanisms of mode I crack growth in FRP have been reviewed by Siddique et al. [11]. They reviewed the main parameters controlling the fracture toughness of such materials. Furthermore, Jia et al. [12] introduced a valuable strategy based on biomimicking to improve the fracture toughness of brittle materials through an intrinsic to extrinsic (ITE) transition. In the ITE transition, toughness started as an intrinsic parameter at the basic material level, but by designing a protein-like effective stress-strain behavior, the toughness at the system level became an extrinsic parameter that increases with the system size with no limit.

The study of translaminar fracture was carried out under the American Society for Testing Materials (ASTM) standard through a single-edge tension specimen. El-Hajjar and Haj-Ali [13] evaluated the applicability of the ASTM E1922 [14] standard for usage with pultruded thick-section composites, which expanded the standard’s scope to encompass these materials. Laffan [15] proposed a study for translaminar tensile failure utilizing a single-edge tension specimen with different thicknesses (by changing the numbers of the plies). He observed that the fracture toughness for specimens increased in thicker 0° plies, and he attributed that to the increase of fiber pull-out. Furthermore, Laffan et al. [16] expressed that using a single-edge tension specimen based on ASTM E1922 was appropriate for quasi-isotropic laminates. However, it rendered less accurate results for extremely orthotropic lay-ups. In the unidirectional fiber polymer composite under pure mode I, according to ASTM E1922, Saadati et al. [17] found that crack propagation started from the notch tip and followed the fiber direction. He et al. [18] demonstrated that brittle matrix could increase fracture work which reduced notch sensitivity. Zappalorto [19] also showed that the shift from notch insensitivity to notch sensitivity was unaffected by the notch root radius, the notch depth, and critical notch size. Moreover, Jamali et al. [20] found crack propagation is perpendicular to the intended development direction under tension.

On the other hand, ASTM D3039 [21] was used to measure the tensile behavior of fiber-reinforced polymers (FRP). Elbadry et al. [22] and El-Wazery et al. [23] studied the tensile behavior of FRP with hand lay-up with different fiber volume fractions, and concluded that increasing the fiber weight contents increased the tensile properties.

The Numerical method is used to simulate processes that cannot be examined or seen in the lab, besides the cost and time of experiments that it saves. Hashin damage model [24,25] provides a good prediction for four modes of failure for matrix and fibers. As Pham et al. [26] demonstrated for the FRP, a discrete crack under a 3D Hashin failure criterion was first used to predict the damage initiation with continuum shell elements. The matrix crack was parallel to the fiber path, and its orientation was determined by the transverse plane’s maximum principal stress, which was validated experimentally. Furthermore, Koloor et al. [27] used Hashin’s modulation by finite element and energy dissipation to regularly get the multidirectional composite material yield value. Duarte et al. [28] simulated the FRP using the ABAQUS program under unidirectional tension based on Hashin’s criteria. They showed that plates with a higher plies number with fibers orientation in the direction of applied load plies had the highest stiffness and strength. They [28] compared the Hashin damage criterion and the eXtended Finite Element Method (XFEM) to predict FRP failure stages. They concluded that the Hashin damage criterion and XFEM predicted the same strength and stiffness of FRP for load levels up to the failure of plies due to matrix cracking. Besides, Contour Integral Method (CIM) is a valid model to estimate J-integrals according to stress intensity factors related to contours above the crack region. El-Sagheer et al. [29] and Abd-Elhady et al. [30,31] used the CIM to determine the stress intensity factor and J-integrals in many applications.

On the other hand, Carpenter et al. [32] confirmed that the matrix under compressive loads follows the relationships established by linear elastic fracture mechanics (LEFM). They found that a log-log plot of the failure load and initial notch length of the experimental data exhibited a linear trend with a slope of −0.54, while the numerical predictions had a linear trend of slope −0.58. As is already known, this plot was expected from LEFM to be linear with a negative slope of 0.5. Liu et al. [33] invoked 3-D finite element analysis (FEA) to study the effect of cohesive zone model (CZM) parameters on the post-buckling and delamination behaviors of FRP under compression. They found that the cohesive strengths mainly affected the unstable delamination stage for the laminates under compression and had little effect on local and global buckling loads. Rozylo [34] concluded that the results obtained based on CZM (numerical) and acoustic emission signals (experimental) showed high agreement. Panettieri et al. [35] used CZM to simulate delaminations growth in compression after impact. Zhou et al. [36] used the shear damage initiation criterion, available in ABAQUS/explicit, to model the shear failure due to fracture within shear bands in metal-ceramic functionally graded bolted joint. It is worth noting that the bolt was made of porous ZrO_2_/(ZrO_2_ + Ni) FGMs. They used Tsai–Wu tensor theory as the failure criteria of the C/SiC plates. They concluded that ZrO_2_ + 15 vol% Ni of two mm thickness is the optimal shear band to balance such bolted joint’s shearing strength and heat insulation performance.

The main objective of the present work is to experimentally study the ability of the ASTM E1922 standard test method to measure the real fracture toughness of unidirectional glass fiber reinforced epoxy (GFRE). It is worth noting that two types of cracked ASTM E1922 specimens have been adopted in the present work. The crack types are single edge through-thickness cracked (TTC) specimen (traditional specimen) and single edge matrix cracked (MC) specimen (suggested in the present work as an unprecedented specimen). The pre-crack exists in the matrix without cutting the fibers behind the crack tip in the MC specimen. In the first stage of the experimental work, MC specimens have been used to show the failure at the loading point. While in the second stage, three types of tensile specimens, namely, smooth, TTC, and MC specimens, have been manufactured with dimensions according to ASTM D3039 to measure the exact value of the fracture toughness of GFRE through the MC ASTM D3039 specimen. No such MC specimen in the fibrous composite is suggested before. Therefore, three-dimensional finite element analysis (3D FEA) simulates the composite laminates based on ASTM E1922 and ASTM D3039 standard tests. Moreover, Hashin criteria and CIM are used in the present simulation to predict the progressive damage and the geometry correction factor (*Y*) of the GFRE cracked specimen, respectively.

## 2. Materials and Methods

### 2.1. Experimental Work

#### 2.1.1. Materials Preparation and Properties

The materials used in the conducted experiment are unidirectional GFRE composites. GFREs are composed of several layers that are bonded together to produce a multilayer composite. The hand lay-up technique is used in the present work. Firstly, glass fibers in the form of roving are impregnated with the resin matrix. The matrix is a high resistance against mechanical stresses and chemical effects and is ready to use after mixing the two components (Resin and Hardener), were supplied by CMB (Giza, Egypt). Continuous glass fibers were supplied by Jushi (Zhejiang, China), 13 μm diameter, acquire the reinforcement.

Consequently, an epoxy-impregnated layer containing continuous unidirectional glass fibers is formed. Then, numerous impregnated layers are stacked on top of one another, and all fibers are aligned in the 0 ° direction (unidirectional). Moreover, glass fibers volume fraction is 37.68%, achieved by ignition loss testing based on ASTM D3171 [37]. Table 1 and Table 2 describe the physical and mechanical properties of fibers and epoxy, respectively, used in the present work.

#### 2.1.2. Specimens Geometry and Experimental Setup

The tested specimens are created by cutting them into pieces from the unidirectional GFRE composite. The Specimens are divided into three types based on the crack type as follows:Smooth Specimen: crack-free specimen as shown in Figure 1a.TTC Specimen: It has an edge crack created using a low-speed saw perpendicular to the fiber direction that left an air gap between two surfaces, as shown in Figure 1b.MC Specimen: the matrix has through crack without cutting the fiber within the crack surfaces, as shown in Figure 1c. The following steps were adopted to create the MC in the unidirectional fiber laminated composites: After placing the dry fibers in each lamina in the hand lay-up technique, the region of MC was defined to put wax around the fibers and drop the Dimethylformamide (DMF), was supplied by Gama Labs (Cairo, Egypt), by syringe. The role of DMF is to stop epoxy curing at the region of MC, and that is producing a debonding region between fibers and epoxy, as shown in Figure 2.

The specimen’s geometric sizes are classified based on ASTM E1922, as shown in Figure 3, and ASTM D3039 Figure 4. The number of layers and the specimen thickness of ASTM E1922 specimens are three and 2.5 mm, respectively. However, the number of layers and specimen thickness of ASTM D3039 specimens are (two and 1.88 mm) and (one and 1.146 mm), respectively. For each GFRE composite type, five specimens are tested for each test to achieve precise results. A universal testing machine (WDW-300), was supplied by Jinan Hensgrand (Jinan, China), conducts the tension test, as shown in Figure 5. To fulfill the load rate requirement for ASTM E1922, i.e., the time from zero to peak-load ranges between 30 and 100 s, a 5 mm/sec loading rate for the ASTM E1922 specimens was used. For ASTM D3039 specimens, a standard head displacement rate of 2 mm/min was used.

### 2.2. Numerical Study

#### 2.2.1. Geometry and Mechanical Properties

The present geometrical model is created as dimensions in experimental work, as shown in Figure 3 and Figure 4. The model has formed from three unidirectional layers of 0° fiber direction perpendicular to the crack surface for ASTM E1922 specimens and two layers for ASTM D3039 specimens. Equations calculate the mechanical properties of glass fiber-epoxy lamina from elasticity approach models [38] as follows:(1)$$E_{11}=V^{F}E^{F}+V^{M}E^{M}+(\frac{\left(4V^{F}V^{M}{\left(\nu^{F}-\nu^{M}\right)}^{2}\right)}{\left(\frac{V^{F}}{K^{M}}+\frac{V^{M}}{K^{F}}+\frac{1}{G^{M}}\right)}),$$
(2)$$\nu_{12}=V^{F}\nu^{F}+V^{M}\nu^{M}+(\frac{\left(V^{F}V^{M}(\nu^{F}-\nu^{M})(\frac{1}{K^{M}}-\frac{1}{K^{F}})\right)}{\left(\frac{1}{G^{M}}\right)}),$$
(3)$$G_{12}=G^{M}(\frac{G^{F}(1+V^{F})+G^{M}V^{M}}{G^{F}V^{M}+G^{M}(1+V^{F})}),$$
(4)$$\mathrm{For}\;G_{23}\;:A{\left(\frac{G_{23}}{G_{M}}\right)}^{2}+2B(\frac{G_{23}}{G_{M}})+C=0,$$
(5)$$\nu_{23}=\frac{K-RG_{23}}{K-RG_{23}},$$
(6)$$E_{22}=2G_{23}(1+\nu_{23}),$$
(7)$$K^{F}=\frac{E^{F}}{2(1-2\nu^{F})(1+\nu^{F})},$$
(8)$$K^{M}=\frac{E^{M}}{2(1-2\nu^{M})(1+\nu^{M})},$$
(9)$$K=\frac{K^{M}V^{M}(K^{F}+G^{M})+K^{F}V^{F}(K^{M}+G^{M})}{V^{M}(K^{F}+G^{M})+V^{F}(K^{M}+G^{M})},$$
(10)$$R=1+4K(\frac{{\left(\nu_{12}\right)}^{2}}{E_{11}})$$

*V* is the volume fraction, *E* is the modulus of elasticity, *G* is the shear modulus, and *ν* is the Poisson’s ratio. *F* is superscript for fiber, and *M* is superscript for matrix. *K^F^*, *K^M^*, and *K* are bulk moduli for fiber, matrix, and composite, respectively, under longitudinal strain. *A*, *B*, and *C* can be obtained from Ref. [38]. Finally, R is a constant.

Table 3 shows the mechanical properties of glass fiber–epoxy lamina, where E_11_ is the longitudinal modulus of elasticity (parallel to fibers direction). E_22_ and E_33_ are the in-plane transverse and out-of-plane moduli of elasticity, respectively. G_12_ and G_13_ are the longitudinal shear moduli in two orthogonal planes containing the fibers, and G_23_ is the out-of-plane shear modulus as well as ν_12_, ν_13_, and ν_23_ are the Poisson’s ratios.

#### 2.2.2. Contact and Boundary Conditions

Axial displacement control was acting on the ASTM D3039 specimens and ASTM E1922 specimens. The ASTM E1922 specimen was loaded from the upper hole in the y-direction and was fixed from the lower hole. The ASTM D3039 specimen was loaded from the upper side in the y-direction and was fixed from the opposite side, as shown in Figure 6. The composite layers are considered perfectly bonded by using the tie contact option in ABAQUS/Standard [39].

#### 2.2.3. Discrete Model

Two element types are used in the present investigation according to the Hashin damage criterion and CIM. Each element in the current model was meshed using continuum shell elements type (SC8R), which confirmed with Hashin damage criterion, and an eight-node linear brick (C3D8R) confirmed with contour integral. The mesh refinement procedure was carried out to ensure that the element’s size does not affect the findings. The composite specimen model’s mesh is shown in Figure 6.

#### 2.2.4. Hashin Damage Model

In the present work, Hashin’s failure criteria are used to predict the translaminar progressive fracture and get the failure load of the present composite specimen according to ASTM D3039 and ASTM E1922. In general, Hashin’s failure criteria are used to predict anisotropic damage in elastic-brittle materials. It is primarily used with FRP and considers four failure modes, namely: fiber breakage in fiber tension, fiber buckling in fiber compression, matrix cracking in matrix tension, and matrix crushing in matrix compression. The initiation criteria have the following general forms:(11)$$\mathrm{Fiber}\;\mathrm{tension}\left(\sigma_{11}\geq \;0\right):F_{f}{}^{t}={\left(\frac{\sigma_{11}}{X^{T}}\right)}^{2}+\alpha {\left(\frac{\tau_{12}}{S^{L}}\right)}^{2},\;$$
(12)$$\mathrm{Fiber}\;\mathrm{compression}\left(\sigma_{11}<0\right):F_{f}{}^{c}={\left(\frac{\sigma_{11}}{X^{C}}\right)}^{2},$$
(13)$$\mathrm{Matrix}\;\mathrm{tension}\;\left(\sigma_{22}\geq 0\right):F_{m}{}^{t}={\left(\frac{\sigma_{22}}{Y^{T}}\right)}^{2}+{\left(\frac{\tau_{12}}{S^{L}}\right)}^{2},$$
(14)$$\mathrm{Matrix}\;\mathrm{compression}\;\left(\sigma_{22}<0\right):F_{m}{}^{c}={\left(\frac{\sigma_{22}}{2S_{T}}\right)}^{2}+\left[{\left(\frac{Y^{C}}{2S^{T}}\right)}^{2}-1\right]\frac{\sigma_{22}}{Y^{C}}+{\left(\frac{\tau_{12}}{S^{T}}\right)}^{2}$$

In the above equations, $X^{T}$ denotes the longitudinal tensile strength, ${\;X}^{C}$ denotes the longitudinal compressive strength, ${\;Y}^{T}$ denotes the transverse tensile strength, ${\;Y}^{C}$ denotes the transverse compressive strength. Furthermore, ${\;S}^{L}$ denotes the longitudinal shear strength, and $S^{T}$ denotes the transverse shear strength. Those can be obtained from Refs. [40,41] and are tabulated in Table 4. α is a coefficient that determines the contribution of the shear stress to the fiber tensile initiation criterion. The $\sigma_{11}{,\;\sigma }_{22},\;$ and $\;\tau_{12}$ components of the effective stress tensor σ. If one of these variables ($F_{F}^{T}$, $F_{F}^{C}$, $F_{M}^{T}$ and $F_{M}^{C}$) reaches the unity, one of the previous failure modes is achieved and the damage process in the elasticity matrix is given by the following equations:(15)$$\left[\begin{array}{c} \sigma_{11}\\ \sigma_{22}\\ \tau_{12} \end{array}\right]=C_{d}\left[\begin{array}{c} \varepsilon_{11}\\ \varepsilon_{22}\\ \varepsilon_{12} \end{array}\right]=\frac{1}{D}\left[\begin{array}{ccc} (1-d_{f})E_{11} & (1-d_{f})(1-d_{m})\nu_{21}E_{11} & 0\\ (1-d_{f})(1-d_{m})\nu_{12}E_{22} & (1-d_{m})E_{22} & 0\\ 0 & 0 & (1-d_{s})G_{12}D \end{array}\right]\left[\begin{array}{c} \varepsilon_{11}\\ \varepsilon_{22}\\ \varepsilon_{12} \end{array}\right],$$
where *C_d_* is the damaged elasticity matrix, and
(16)$$D=1-(1-d_{f})(1-d_{m})\nu_{12}\nu_{21},$$
(17)$$d_{f}=\{\begin{array}{c} d_{f}^{t}\\ d_{f}^{c} \end{array}\begin{array}{c} ,\\ , \end{array}\begin{array}{c} if\\ if \end{array}\begin{array}{c} \left(\sigma_{11}\geq 0\right)\\ \left(\sigma_{11}<0\right) \end{array},$$
(18)$$d_{m}=\{\begin{array}{c} d_{m}^{t}\\ d_{m}^{c} \end{array}\begin{array}{c} ,\\ , \end{array}\begin{array}{c} if\\ if \end{array}\begin{array}{c} \left(\sigma_{22}\geq 0\right)\\ \left(\sigma_{22}<0\right) \end{array},$$
(19)$$d_{s}=1-(1-d_{f}^{t})(1-d_{f}^{c})(1-d_{m}^{t})(1-d_{m}^{c})$$
where *d_f_*, *d_m_*, and *d_s_* are the damage variables for fibers, matrix, and shear, and they can be obtained from Ref. [28]. The initial values of *d_f_*, *d_m_*, and *d_s_* are assumed to be zero and gradually increased until they reach the unity. According to equation 15, as the load increases, the effective strength of the elements decreases. Elements with a damage variable equal to the unity are excluded because they can no longer withstand such stress.

#### 2.2.5. Contour Integral Method

Contour Integral provides a standard scheme of finite elements, usually helping to match the mesh to the cracked frame, specifically describing the crack front Assessment can be viewed as a simulated displacement of a material block across the crack tip in 3D, represented by contours where each contour is a circle of fully surrendered components. Each contour gives an integral assessment of the contours. In the present work, CIM was used to obtain the stress intensity factor (SIF) in mode I (*K_I_*) for GFREs for TTC and MC specimens.

In the TTC specimen, the crack cuts the matrix and fiber as one bulk material without slippage between the matrix and the fiber. And the CIM can be used to get the SIF directly. Otherwise, for MC-specimen, the crack at matrix only; therefore, the matrix is slipping on the fiber during the tensile load. A unique and simple method is used to simulate the MC specimen to get the SIF. As is already known, the bond strength between fiber and matrix can be classified into three cases. Namely, perfect bond, i.e., no fiber bridging, and the composite failed as a monolithic material (unpreferable case), strong bond (preferable case), i.e., fiber bridging achieved with different efficiency depending on the debonding length of the fiber (Notch sensitivity achieved), as shown in Figure 7.

The last case is a weak bond, i.e., the crack grows parallel to the applied load and fiber direction (notch sensitivity does not achieve). MC model is utilized to calculate K_I_ under debonding zones between fibers and epoxy. To simulate the fiber bridging simply, the debonding zones translated into degrees as *θ* is debonding angle in degree between fibers and matrix in MC model. Figure 7 shows a simulation of the deboning zone between fibers and matrix. The fibers in the debonding zone act as a spring with efficiency depending on its stiffness (Modulus of elasticity of fibers) and length (debonding length of fibers). Different values of *θ* are adopted; namely, 0, 10, 20, 30, 40, 50, 60, 70, 80, 90, 100, 110, and 120, to study the effect of debonding length on the crack closure, i.e., the reduction in SIF.

## 3. Results and Discussions

### 3.1. Progressive Failure

#### 3.1.1. Progressive Failure for ASTM E1922 Specimen

Figure 8 shows the photographs of the progressive failure for ASTM E1922 specimens under the tensile load through the two holes. For the TTC specimen, the crack started its propagation perpendicular to the direction of the fibers up to about 1 mm, as shown in Figure 8a. By definition, the fracture toughness can be calculated at the crack initiation load. After that, this propagating crack changed its direction to be parallel to the applied load and fibers direction until final failure, as shown in Figure 8a. This observation is confirmed with results obtained by Ref. [17]. On the other hand, smooth specimen and MC specimen are failed through the loaded holes instead of the middle of cross-section, as illustrated in Figure 8b. It can be concluded that ASTM E1922 only works with TTC specimens and does not work with MC specimens. To avoid failure at the point of loadings in the ASTM E1922 specimens, MC specimens should be subjected to uniform tensile loads as followed in the tensile test in ASTM D3039. Therefore, the fracture toughness of fibrous composite materials has been measured through TTC and MC ASTM D3039 specimens.

#### 3.1.2. Progressive Failure for ASTM D3039 Specimen

Based on ASTM D3039, a flat specimen setup is conducted experimentally for smooth and cracked specimens to examine the tensile and fracture behaviors of GFRE material. The dimensions of TTC and MC specimens followed ASTM D3039 with a/W = 0.5. The progressive damage of the smooth specimen under the tensile load can be depicted in Figure 9, divided into major steps in both fibers and matrix damage. In the first step at displacement = 5.531 mm, a longitudinal crack propagates in one side of the specimen, the matrix has started to be damaged, and fibers are sustaining the load. In the second step, at load 6.914 mm, two longitudinal cracks propagate on opposite sides, the matrix has partly been damaged, and fibers are standstill sustaining the load. Finally, at load-displacement 7.202 mm, the matrix has been damaged and splitting from the fibers, while fibers are partly damaged.

Figure 10 shows several major steps in both fibers and matrix damages in the case of TTC specimens. Firstly, at load-displacement at 1.63 mm, a longitudinal crack is propagating besides the crack-tip side, the matrix has started to be damaged, and fibers are sustaining the load. At applied displacement at 4.61 mm, a longitudinal crack is still propagating toward grip, and the matrix has partly been damaged. The fibers partly sustain the load at applied displacement 5.33 mm, and the matrix has been damaged, splitting from the fibers, while fibers are partly damaged.

Hashin’s failure criteria were executed in the present work to predict the fiber and matrix damages due to tensile or compression stress, as shown in Figure 9 and Figure 10. From Figure 9, it was observed that the matrix is beginning damage in the specimen’s middle after that; the debonding between the matrix and fiber has happened. This debonding and matrix damage grows; finally, the fibers are damaged in the upper and lower part of the specimen. From Figure 10, it can be seen that the damage of matrix and fiber are beginning from the pre-crack tip and is growing in the fiber direction parallel to the applied load that confirms with [17,42]. From Figure 9 and Figure 10, the Hanshin model can simulate the progressive damage of smooth and TTC ASTM D3039 specimens.

From Figure 9 and Figure 10 (experimental and numerical results), it can be concluded that the matrix in the ASTM D3039 smooth and TTC specimens are completely damaged and split from the fibers into all layers. After that, the fibers sustain the load, and the main damage type concerns the matrix in tension load. That is why the FRP matrix carries the major load.

The main conclusion from the final failure shape of the TTC specimen of ASTM E1922 and ASTM D3039 is that the ASTM E1922 is completely fractured into two pieces, while in the ASTM D3039, the specimen is completely damage but not separate.

As shown in Figure 11, the debonding between the epoxy-matrix and glass-fiber in the final failure shape of MC specimen of ASTM D3039 is much more observed than smooth and TTC specimens. Furthermore, the pre-crack in the TTC specimen becomes blunting, and the debonding between the fiber and matrix begins from the crack tip then grows perpendicular to the pre-crack.

### 3.2. Load Displacement Curve

Figure 12 shows the load-displacement curve of ASTM E1922 and ASTM D3039 specimens under tensile load. It can be observed that the applied load increase to reach a peak value which the matrix is completely damaged (as shown in Figure 8, Figure 9 and Figure 10), then the applied load is suddenly dropped. As expected, the applied load peak value for the smooth specimen is more than the MC and TTC specimens of ASTM D3039. Moreover, the value of the peak load of the TTC specimens is lower than that of MC specimens. For the same a/W, the peak value of the applied load of the TTC specimen of ASTM D3039 (7.11 kN) is much higher than ASTM E1922 (2.24 kN), although the thickness of the specimen of ASTM E1922 (2.5 mm) is higher than that of ASTM D3039 (1.88 mm). This difference may be attributed to the applied load being eccentric in ASTM E1922, while the applied load is uniform distributed in ASTM D3039.

The numerical results of the Hashin model can be verified by comparing it with the experimental result for smooth and TTC ASTM D3039 specimens in Table 5. Table 5 shows a good agreement between the peak applied load values of smooth and TTC specimens, computed from the numerical and experimental methods. Therefore, Hashin numerical method can be used to determine the damage values of composite laminates. It is worth noting that the load at which the crack initiated from the notch root of the TTC ASTM E1922 specimen is indicated in Figure 12.

The thickness effect was also studied by changing the number of layers on the tensile and fracture behavior of smooth, MC, and TTC specimens of ASTM D3039. Two specimen thicknesses are 1.146 mm and 1.88 mm that formed using one and two layers, respectively. The thickness or the number of layers does not affect the mode of failure of the smooth, MC, and TTC specimens of ASTM D3039. The pop-in events can be expressed as the damage growth in the matrix, showing a temporary drop in load, and then the fibers sustain the load again before the matrix is completely damaged. Figure 13a. shows that the modulus of elasticity (*E*) and the ultimate tensile strength (UTS) of unidirectional GFRE of two layers (1.88 mm) are higher than those of GFRE of on layer (1.46 mm). In the case of two layers, the initial tangent modulus *E_in_* is about 33 GPa, and this slope decreased at stress equals about 150 MPa to be 20 GPa, while the UTS equals 466 MPa at strain equals 0.02. In one layer, the *E_in_* is about 19 GPa, and this slope decreased at stress equals about 270 MPa to be 10 GPa, while the UTS equals 409 MPa at strain equals 0.027. Although GFRE with two layers has higher mechanical properties, the deviation of *E* in GFRE with one layer occurs at stress higher than that of GFRE with two layers. Furthermore, the maximum stress of GFRE with two layers is higher than that of GFRE with one layer either for MC specimen or for TTC specimen, as shown in Figure 13b,c, respectively. It is worth noting that the load at which the crack initiated from the notch root is indicated in Figure 13b,c.

### 3.3. The stress Concentrator Factor (SCF)

As mentioned before, the unidirectional GFRE experienced initial notch sensitivity behavior followed by notch insensitivity. Therefore, the cracked specimens of ASTM D3039 were analyzed based on the first principle of mechanics of materials, i.e., SCF, as listed in Table 6. The experimental SCF (*K_t_^Experimental^*) is calculated as the ratio of the peak load of the smooth specimen divided by the peak load of the notched (cracked) specimen, while the analytical SCF (*K_t_^Analytical^*) is calculated based on the following Equation:(20)$${K_{t}}^{\mathrm{Analytical}}=(1+2\sqrt{\frac{D}{\rho }})$$

*D* (7.5 mm) and *ρ* (0.5) are the depth and radius of the notch, respectively.

It is clear from Table 6 that the coefficient of variation (CV) of smooth specimens is lower than that of TTC specimens. This difference may be attributed to the precision of notch radius manufacturing. *K_t_^Experimental^* for the one-layer specimen is higher than that of two-layer specimens, while the ratios of both values to those calculated analytically, *K_t_^Analytical^*, are lower than the quarter. Based on these ratios, it can be concluded that the study of the notch sensitivity by the SCF is acceptable.

Furthermore, *K_t_^Experimental^* is around two (2.12 and 1.91 for both cases), and the ratio of notch depth to specimen width is 0.5, i.e., the net stress of notched specimen is twice its gross stress. Comparing the value of this ratio with the value of *K_t_^Experimental^* reveals that both values are more or less the same. The net stress analysis is evidence to prove the notch insensitivity of the present unidirectional laminated GFRE.

### 3.4. Fracture Toughness

#### 3.4.1. The Stress Intensity Factor for TTC Specimen

The present work uses the CIM to calculate the SIF for TTC of ASTM E1922 and ASTM D3039. In the beginning, it must be verifying the numerical results of SIF values. To verify the present numerical results, a comparison between the SIF values extracted from the CIM and the SIF and geometry correction factor which extracted from the following analytical Equation:

For TTC ASTM E1922 specimens:(21)$$K_{1}=\frac{1}{{\left(1-(\frac{a}{w})\right)}^{3/2}}((\frac{P}{\left(BW^{1/2}\right)})(1.4+(\frac{a}{w})){\left(\frac{a}{w}\right)}^{1/2}(3.97-10.88(\frac{a}{w})+26.25{\left(\frac{a}{w}\right)}^{2}-38.9{\left(\frac{a}{w}\right)}^{3}+30.15{\left(\frac{a}{w}\right)}^{4}-9.27{\left(\frac{a}{w}\right)}^{5})),$$
where:(22)$$Y=\frac{K_{I}}{\sigma \;\sqrt{\pi a}}=\sqrt{\frac{1}{\pi }}\;[1.4+\alpha ][3.97\;-\;10.88\alpha +26{.25\alpha }^{2}\;-\;38{.9\alpha }^{3}+30{.15\alpha }^{4}\;-\;9{.27\alpha }^{5}{]/[1\;-\;\alpha ]}^{1.5}$$
where *K_I_* is SIF in (MPa·√m), *P* is Load in (MN), *B* is the thickness of the specimen in (m), *W* is the width of the specimen in (m), *a* is the crack length in (m), *Y* is geometry correction factor in (dimensionless), and α is (*a/W*).

Figure 14 illustrates the comparison between the theoretical results of SIF and Y, which are computed from Equations (21) and (22) of TTC ASTM E1922 for different crack length ratios to confirm that the model works well (*a/W* = 0, 0.2, 0.3, 0.4. 0.5, and 0.6) and the corresponding numerical results. From Figure 14, Table 7 and Table 8, it can be seen that there is a good agreement between the numerical and analytical results. Table 7 shows the absolute error percentage for each crack length, and the absolute error values are less than 4%.

For TTC ASTM D3039 flat specimens, the following Equation from Ref. [43] can be used:(23)$$K_{I}=Y\sigma \sqrt{\pi a},$$
where:(24)$$Y=\frac{1}{\sqrt{\pi }}(1.99-0.41(\frac{a}{w})+18.7{\left(\frac{a}{w}\right)}^{2}-38.48{\left(\frac{a}{w}\right)}^{3}+53.85{\left(\frac{a}{w}\right)}^{4})$$

The numerical model using the CIM is shown in Table 9. As illustrated in Table 9, the error is less than 0.9%, indicating a good agreement between the theoretical and numerical results of the SIF of the TTC ASTM D3039 specimen.

#### 3.4.2. The Stress Intensity Factor for MC Specimen

To the best of the authors’ knowledge, no analytical formula in the literature can predict the SIF or *Y* for ASTM E1922 and ASTM D3039 MC-specimen. The contour integral model modifications are made to calculate *Y* for MC specimens for a different debonding length of fibers due to the difficulty of predicting this length. Figure 15 shows the relation between the *Y* at *a/W* = 0.5 against the debonding angle (*θ*) for ASTM E1922 and ASTM D3039. In general, *Y* and subsequently SIF increased by increasing *θ*, i.e., increasing the debonding length of the fibers. As it is already known, the efficiency of fiber bridging decreased by increasing this length, see Figure 7. The results show that the fibers bridging phenomenon works very well to exercise crack closure at the small values *θ*, i.e., the debonding length in the intact fibers behind the crack tip is short. At the large values *θ*, the closing of the fibers behind the crack tip decreased due to the increase in debonding length, and subsequently, the *Y* increased rapidly.

The lower bound of *Y* (<<1) at *θ* = 0° does not take place (perfect bond), and the composite failed as monolithic materials. This hypothesis is supported by the experimental results of Mode-I fracture toughness of Carbon-Carbon composites [44]. Kumar et al. [44] concluded that their fractured surface clearly showed brittle fracture behavior, and the fracture may be explained by energy-consuming mechanisms like fiber pull-out and fiber debonding. However, this brittle fracture behavior is evidence for the absence of fiber bridging. The fiber bridging is the main reason for converting the brittle failure of brittle composites to ductile failure. As mentioned before, the good bond between the fiber and the matrix is favorable to ensure the ductile failure, not perfect bond, not weak bond; both cause brittle failure. Figure 15 shows that *Y* becomes higher than the unity at a certain value *θ*, depending on the specimen type. The values of *Y* below the unity mean that the SIFs of MC cracked specimens are lower than that of the corresponding TTC specimens with infinite width. The upper bound *Y* (the effect of fiber bridging is diminished) must not exceed the corresponding value of *Y* for the TTC specimen, as stated in Figure 15. For a deeper understanding, supposing MC and TTC specimens with the same geometry and a/W are subjected to the same stress. According to Equation (23) and the values of *Y* in Table 10, the SIF of the MC specimen is much lower than that of the TTC specimen due to the crack closure achieved by fiber bridging. The closing SIF due to fiber bridging can be calculated by subtracting the SIF of the MC specimen from the SIF of the TTC specimen (SIF_FB_ = SIF_TTC_ –SIF_MC_). Furthermore, the closing SIF decreased by increasing *θ*, and it is equal to zero when the fiber bridging diminished.

#### 3.4.3. The Apparent Fracture Toughness

The apparent fracture toughness (*K_Q_*) of TTC and MC ASTM E1922 and ASTM D3039 specimens are calculated based on the values of crack initiation loads indicated in Figure 12 and Figure 13 and listed in Table 10. Equations (21)–(24) are used to predict the values of apparent fracture toughness of TTC ASTM E1922 and ASTM D3039 specimens, while Figure 15 is used to predict the values of *Y* for MC ASTM D3039 specimens and substituted these values in Equation (23). It is worth noting that the apparent fracture toughness of MC ASTM D3039 specimens must be calculated using the appropriate value *θ* (*θ* = 80° in the present work to be *Y* of MC specimen is equal to 0.62 times that of TTC specimen, see Table 10. As already known, the crack initiation load decreases with increasing *θ*; thus, it is wrong to study the change in the *θ* values with the same crack initiation load because it would give a misleading trend that the apparent fracture toughness increases with increasing *θ*.

As expected, the *K_Q_* of TTC ASTM D3039 is lower than that of MC ASTM D3039, although *Y* of the MC specimen is much lower than that of the TTC specimen. It is clear from Table 10 that the *K_Q_* measured based on TTC ASTM D3039 is about 1.45 times that of TTC ASTM E1922. Furthermore, the *K_Q_* of TTC ASTM D3039 is not affected by the number of layers, while the *K_Q_* of MC ASTM D3039 is affected by the number of layers. The *K_Q_* of the MC ASTM D3039 specimen with two layers is about 1.3 times that of the MC ASTM D3039 specimen with one layer so that the ratio of the MC ASTM D3039 specimen with two layers to the corresponding TTC ASTM D3039 specimen (83.33/51.13 = 1.62) is higher than the ratio for one-layer specimens (62.72/50.7 = 1.24).

### 3.5. The Crack Mouth Opening Displacement, CMOD

Figure 16 shows the crack mouth opening displacement, CMOD, versus the applied load for TTC specimen of ASTM E1922 and ASTM D3039. The values of CMOD increase by increasing the applied load to reach the peak value, then it drops; this trend is similar for the two specimens. From Figure 16, it can be seen that the shapes of the curves of applied load versus CMOD and the applied load versus displacement are similar.

## 4. Conclusions

Based on the present experimental and numerical results, it can be concluded that the TTC ASTM E1922 specimen is not an appropriate candidate to get the fracture toughness of laminated fibrous composite materials. The suggested MC specimen is a good candidate to predict the actual fracture toughness of such materials. The MC ASTM E1922 specimen failed to initiate the crack from the root of the pre-notch, and the failure occurred at the point of loading due to the presence of stress concentration. Single edge cracked specimen with *a/W* = 0.5 under uniform loading is suggested to predict the fracture toughness. The dimensions of this specimen followed ASTM D3039 specimen; both TTC and MC ASTM D3039 specimens show notch sensitivity, i.e., matrix crack growth perpendicular to the fiber direction up to crack length equals about 1 mm. After that, the crack kinks to grow parallel to the fiber direction, i.e., notch insensitivity. It is suggested that the fracture toughness might be calculated before the occurrence of kinked crack.

The present experimental and numerical results reveal several points of interest: (1) The SCF is a good candidate to predict notch sensitivity in such composites. (2) The *K_Q_* measured based on TTC ASTM D3039 is about 1.45 times that of TTC ASTM E1922. (3) In the case of MC specimens, the numerical results show that *Y* and subsequently SIF increased by increasing the debonding length of the fibers. (4) As expected, the ratio of the fracture toughness measured from the MC specimen to that measured from the TTC specimen is greater than the unity, ranging from 1.24 in the one-layer specimen to 1.62 in the two-layer specimen. (5) The *K_Q_* measured from MC ASTM D3039 increased with increasing the number of layers.

## Figures and Tables

**Figure 1 polymers-13-03129-f001:**
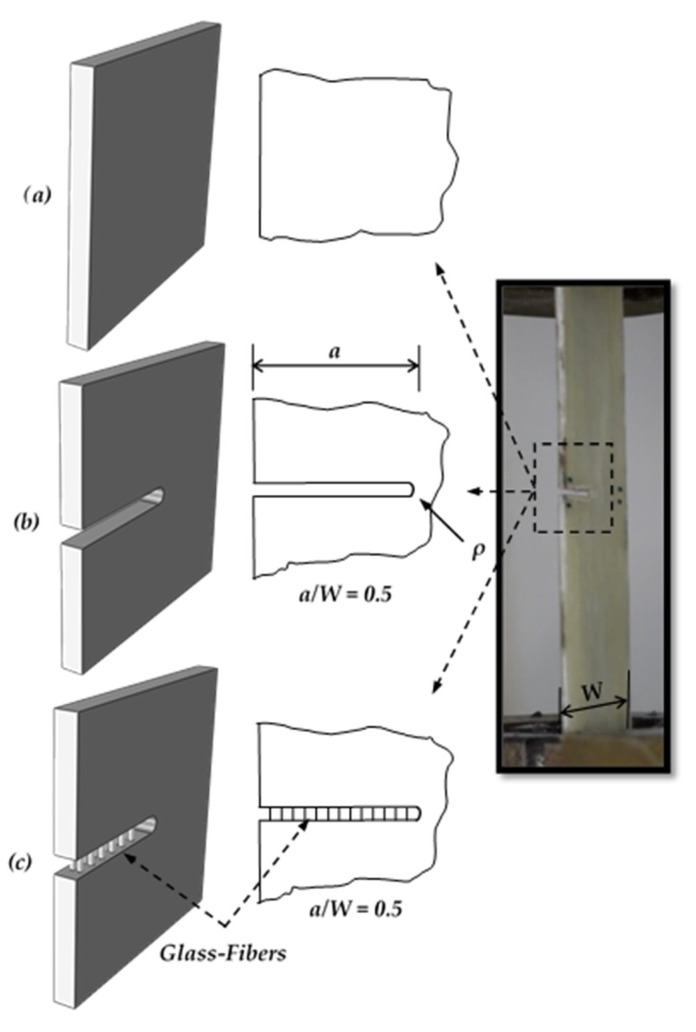
Types of cracks, (**a**) smooth (crack free), (**b**) through thickness crack, and (**c**) matrix crack.

**Figure 2 polymers-13-03129-f002:**
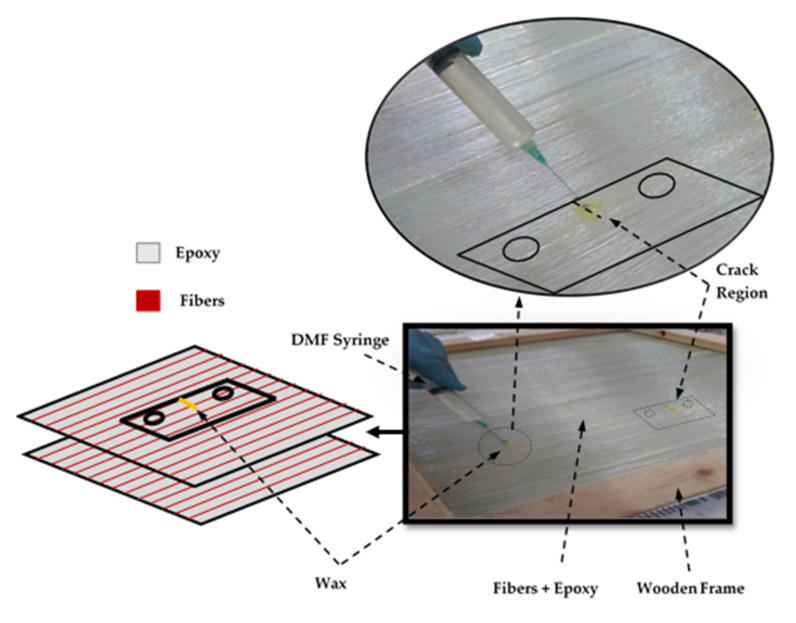
Preparation of matrix crack by wax and DMF.

**Figure 3 polymers-13-03129-f003:**
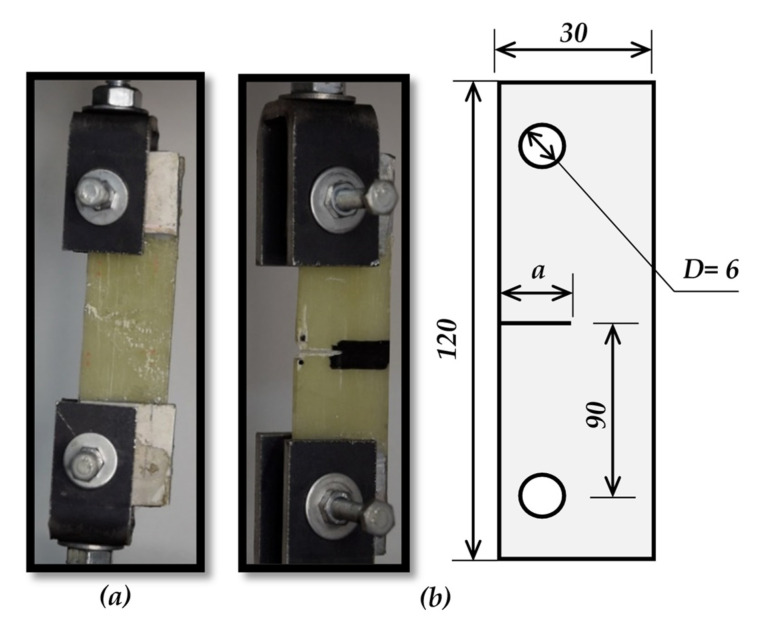
The geometry of ASTM E1922 specimens, (**a**) smooth specimen, and (**b**) TTC Specimen. All dimensions in mm.

**Figure 4 polymers-13-03129-f004:**
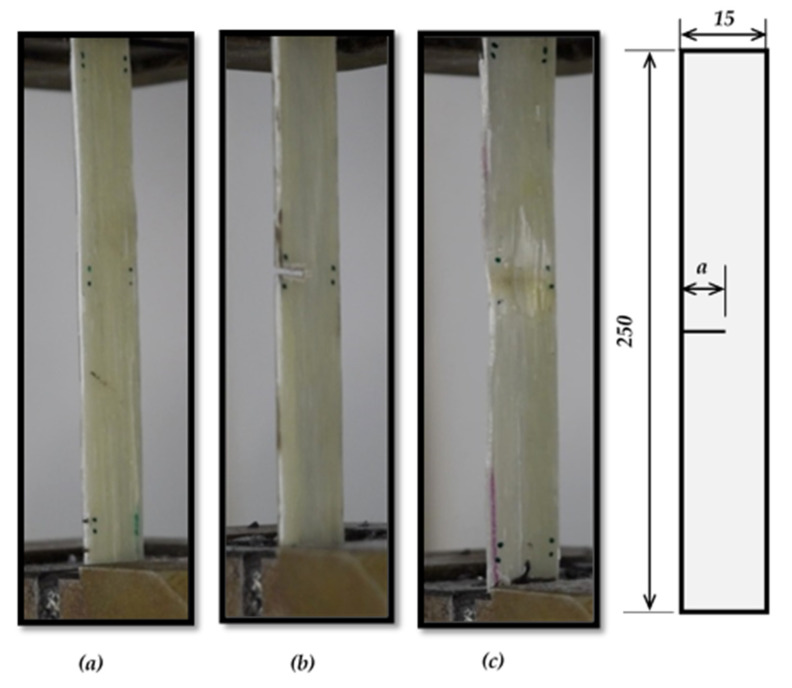
The geometry of ASTM D3039 specimens, (**a**) smooth specimen, (**b**) TTC specimen, and (**c**) MC specimen. All dimensions in mm.

**Figure 5 polymers-13-03129-f005:**
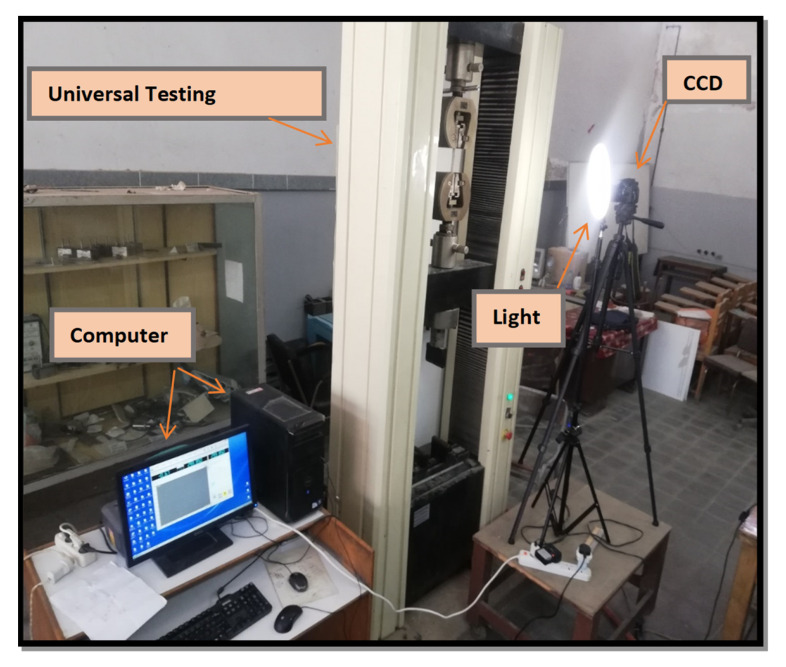
Experimental Setup.

**Figure 6 polymers-13-03129-f006:**
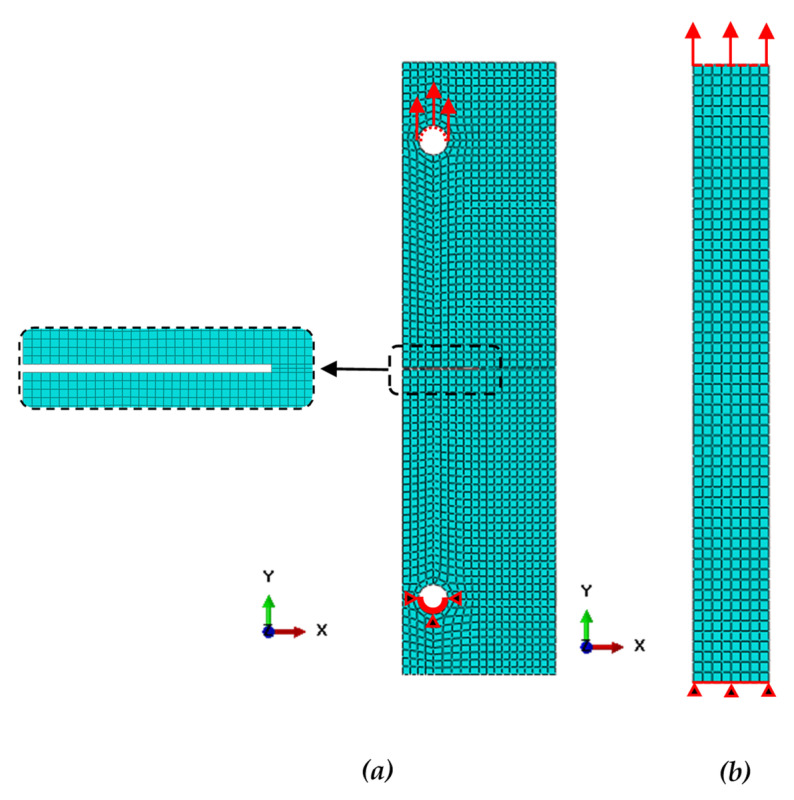
Mesh and boundary conditions of the present model, (**a**) ASTM E1922, and (**b**) ASTM D3039.

**Figure 7 polymers-13-03129-f007:**
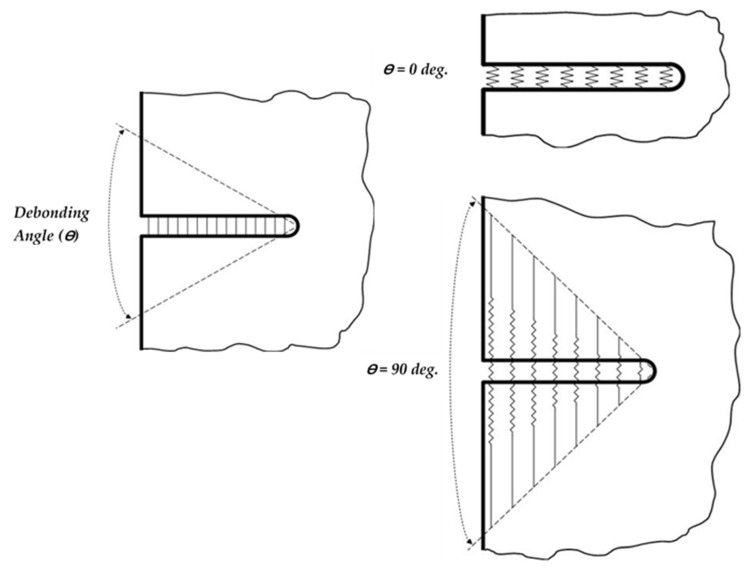
A schematic diagram showing the simulation of debonding zone between fibers and matrix in MC Model.

**Figure 8 polymers-13-03129-f008:**
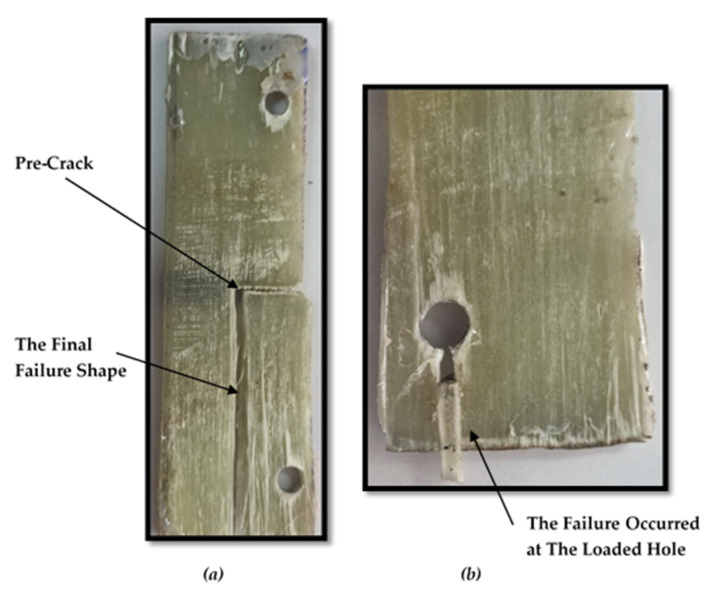
Photographs showing the progressive damage for (**a**) TTC ASTM E1922 specimen and (**b**) MC ASTM E1922 specimen.

**Figure 9 polymers-13-03129-f009:**
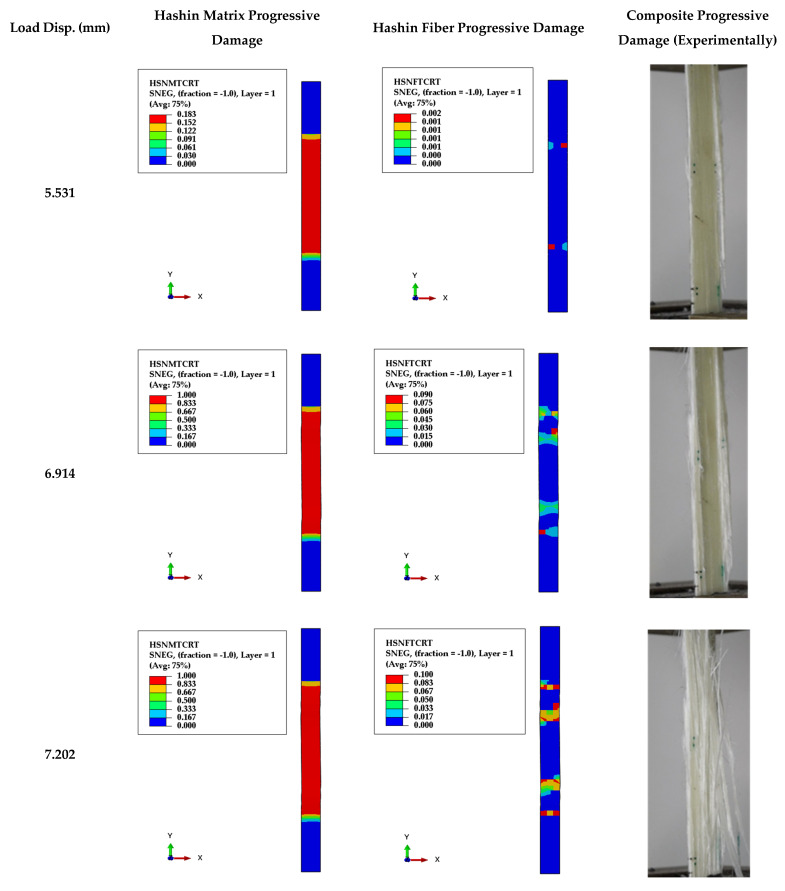
Numerical and experimental progressive damage failure of ASTM D3039 specimen.

**Figure 10 polymers-13-03129-f010:**
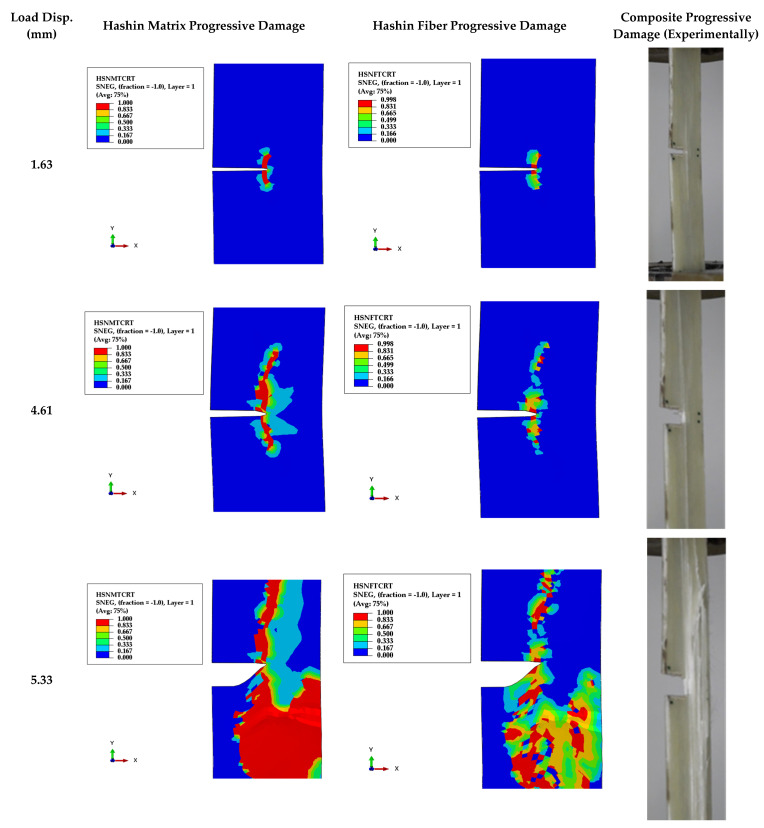
Numerical and experimental progressive damage failure of TTC ASTM D3039 specimen.

**Figure 11 polymers-13-03129-f011:**
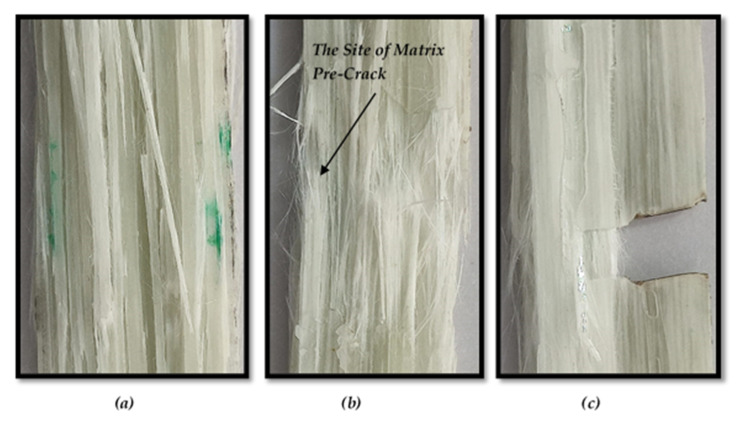
Close-up view of experimental final failure shape of ASTM D3039 for (**a**) Smooth specimen, (**b**) MC specimen, and (**c**) TTC specimen.

**Figure 12 polymers-13-03129-f012:**
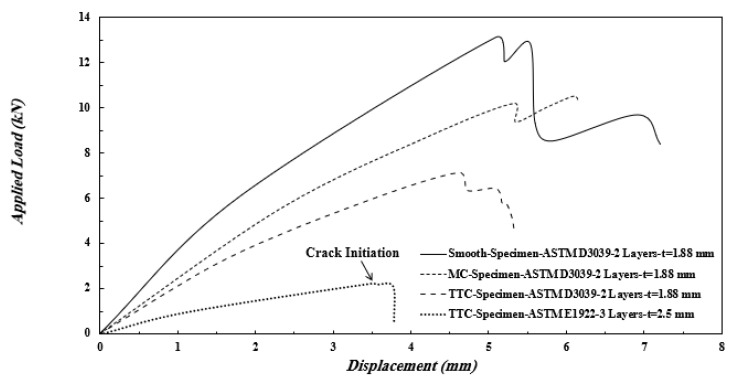
Load-Displacement diagram for ASTM E1922 and ASTM D3039 specimens.

**Figure 13 polymers-13-03129-f013:**
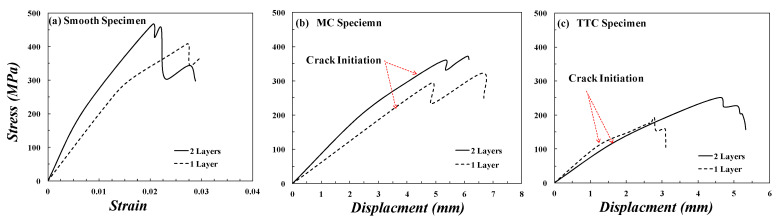
Specimens of ASTM D3039 made of GFRE with one layer and two layers: (**a**) Stress-Strain curve of the smooth specimen, and Stress-Displacement diagram of (**b**) MC specimen, and (**c**) TTC specimen.

**Figure 14 polymers-13-03129-f014:**
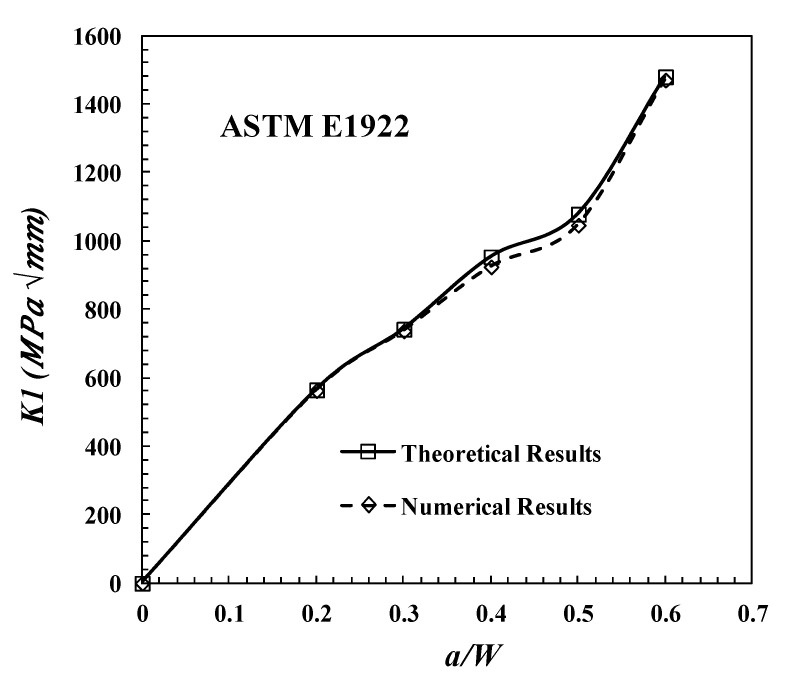
Comparison between the theoretical and numerical results of SIF for TTC ASTM E1922 specimen.

**Figure 15 polymers-13-03129-f015:**
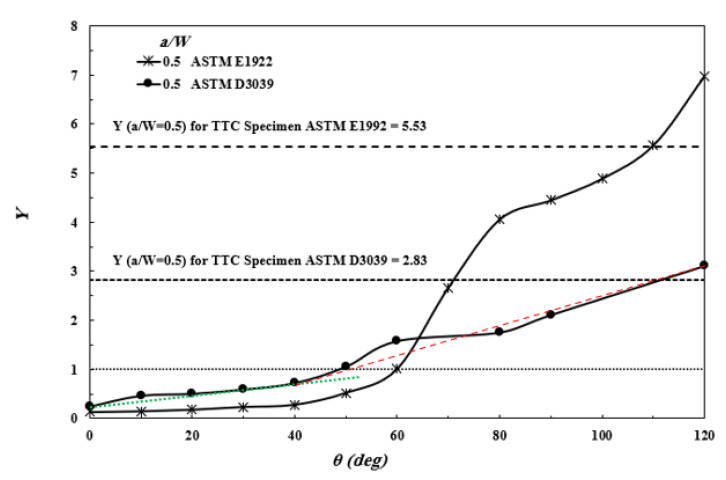
The geometry correction factor (*Y*) of MC ASTM E1922 and ASTM D3039 specimens with *a/W* = 0.5 and different debonding angles (*θ*).

**Figure 16 polymers-13-03129-f016:**
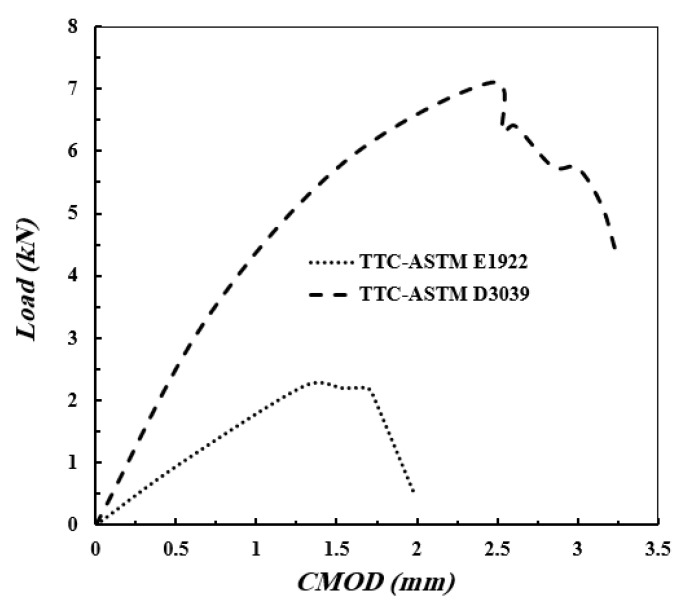
Crack mouth opening displacement versus the applied load of TTC ASTM E1922 and ASTM D3039 specimens.

**Table 1 polymers-13-03129-t001:** The physical properties of the used materials.

Material	Physical Properties	Value
Epoxy	Density	1.11 ± 0.02 Kg/L
	Abrasion resistance	1–6 cm^3^/50 cm^2^
	Temperature resistance	humid 90 °C/dry 140 °C
E-glass Fiber	Filament diameter	13 µm
	Linear density	2400 tex

**Table 2 polymers-13-03129-t002:** The mechanical properties of E-glass fiber and epoxy resin.

Mechanical Properties	E-Glass Fiber	Epoxy
Modulus of elasticity, *E*, (GPa)	82	3.24
In-plane shear modulus, *G*, (GPa)	30.13	1.1912
Poisson’s ratio, *ν*,	0.22	0.36

**Table 3 polymers-13-03129-t003:** The mechanical properties of E-glass fiber/epoxy composite.

	E_11_ (GPa)	E_22_ (GPa)	E_33_ (GPa)	ν_12_	ν_13_	ν_23_	G_12_ (GPa)	G_13_ (GPa)	G_23_ (GPa)
Glass fiber/epoxy	32.9	6.7	6.7	0.29	0.29	0.49	2.46	2.46	2.25

**Table 4 polymers-13-03129-t004:** Hashin Model Constants.

	$X^{T}\;(\mathrm{MPa})$	$X^{C}\;(\mathrm{MPa})$	$Y^{T}\;(\mathrm{MPa})$	$Y^{C}\;(\mathrm{MPa})$	$S^{L}\;(\mathrm{MPa})$	$S^{T}\;(\mathrm{MPa})$
Glass fiber/epoxy	991.9	502.7	548.3	53	65	26.5

**Table 5 polymers-13-03129-t005:** The comparison between the experimental and numerical results of peak load.

	Smooth Specimen Peak Load in (kN)	Through Thickness Crack Specimen Peak Load in (kN)
Experimental	12.86	7.11
Hashin Damage Model by ABAQUS	12.78	6.93

**Table 6 polymers-13-03129-t006:** The experimental and numerical stress concentrator factors (*K_t_^Experimental^*) & (*K_t_^Analytical^*), respectively, for TTC specimens of ASTM D3039.

	1-Layer-Specimen		2-Layers-Specimen	
	Smooth	TTC	Smooth	TTC
Avg. Peak Load (kN)	7.36	3.47	13.12	6.87
Standard Deviation (kN)	1.2	0.8	1.79	1.6
coefficient of variation, CV	0.16	0.23	0.14	0.23
*K_t_^Experimental^*		2.12		1.91
*K_t_^Analytical^*		8.75		8.75

**Table 7 polymers-13-03129-t007:** Comparison between the theoretical and numerical results of SIF and *Y*.

*a/w*	*K_I_* (MPa. √mm)—ASTM E1922 Specimen		% Error
	Theoretical	Numerical	
0.2	565.16	563.01	0.38
0.3	742.29	737.29	0.67
0.4	953.35	924.52	3.02
0.5	1078.58	1046.35	2.98
0.6	1479.89	1470.31	0.64

**Table 8 polymers-13-03129-t008:** Comparison between the theoretical and numerical results of *Y* for TTC ASTM E1922 specimen.

*a/w*	Geomertry Correction Factor (*Y*) —ASTM E1922 Specimen	
	Theoretical	Numerical
0.2	3.253235403	3.094618148
0.3	3.66913351	3.433058442
0.4	4.382919067	4.726436044
0.5	5.533784816	5.348836997
0.6	7.478289346	7.640451598

**Table 9 polymers-13-03129-t009:** The stress intensity factor (K_1_) and the geometry correction factor (*Y*) for TTC specimens as (*a/W* = 0.5).

	*K_I_* (MPa·√mm)—ASTM D3039 Specimen	*Y*
Theoretical	1701.35	2.83
Numerical	1715.312	2.85

**Table 10 polymers-13-03129-t010:** The apparent fracture toughness (*K_Q_*) and the geometry correction factor (*Y*) for TTC specimens with (*a/W* = 0.5).

***a/W* = 0.5**	**ASTM E1922—TTC Specimen**	**ASTM D3039—TTC Specimen**	
	**3-Layers**	**1-Layer**	**2-Layers**
Load at crack initiation (kN)	2.2	2.101	3.32
*Y*	5.53	2.83	2.83
*K_Q_* (MPa·√m)	35.23	50.7	51.13
***a/W* = 0.5**	**ASTM D3039—MC Specimen**	**ASTM D3039—MC Specimen**	
	**1-Layer**	**2-Layer**	
	***θ* = 80°**	***θ* = 80°**	
Load at crack initiation (kN)	4.18	8.7	
*Y*	1.76	1.76	
*K_Q_* (MPa·√m)	62.72	83.33	

## Data Availability

The data presented in this study are available on request from the Corresponding author.

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
