# Peer review of "An Assessment of ASTM E1922 for Measuring the Translaminar Fracture Toughness of Laminated Polymer Matrix Composite Materials"

_polymers, 2021, doi:10.3390/polym13183129_

Round 1

Reviewer 1 Report

The present manuscript entitled ‘An assessment of ASTM E1922 for measuring the Translaminar Fracture Toughness of Laminated Polymer Matrix Composite Materials” authored by Islam El-Sagheer et al. describes the current study is to overcome this deficiency by suggesting unprecedented cracked specimens, i.e., matrix cracked (MC) specimens. The manuscript is well organized, and the study is sufficiently performed. The result analysis is very accurate and adequate, lacks major errors. Therefore, I would recommend the publication of the manuscript in the “Polymers” after some MINOR improvements.

I advise the authors to take the following points into account while revising their manuscript.

Comment 1:  Firstly, I would like to draw the attention of the authors that I found there are so many typographical errors in the manuscript, so authors need to correct them in the revised manuscript. For e.g. In the table, “1 – 6 cm3/50 cm2” should be “1, 1 – 6 cm3/50 cm2” etc.,

Comment 2:  Define ASTM in the abstract.

Comment 3:  Some relevant references in this area are still missing in the introduction section, so include some significant relevant references from recent years. Moderate

Comment 4: Moderate English changes are required in the manuscript.

Comment 5: The provided Figure's quality is very poor. So, provide the high-resolution Figures.

Comment 6: The text is overlapping in Table 10, arrange them properly.

Author Response

Response to Reviewer 1 Comments

General Comment: The present manuscript entitled ‘An assessment of ASTM E1922 for measuring the Translaminar Fracture Toughness of Laminated Polymer Matrix Composite Materials authored by Islam El-Sagheer et al. describes the current study is to overcome this deficiency by suggesting unprecedented cracked specimens, i.e., matrix cracked (MC) specimens. The manuscript is well organized, and the study is sufficiently performed. The result analysis is very accurate and adequate, lacks major errors. Therefore, I would recommend the publication of the manuscript in the “Polymers” after some MINOR improvements.

Response: The authors would like to thank the reviewer for this comment.

I advise the authors to take the following points into account while revising their manuscript.

Point 1: Firstly, I would like to draw the attention of the authors that I found there are so many typographical errors in the manuscript, so authors need to correct them in the revised manuscript. For e.g. In the table, “1 – 6 cm3/50 cm2” should be “1, 1 – 6 cm3 /50 cm2 etc.,

Response 1: Careful revisions have been made to correct such typographical errors. Besides, Table 1 has been modified as follows:

 Table 1. The physical properties of the used materials.

Material

Physical Properties

Value

Epoxy

Density

1.11 ±0.02 Kg/l

Abrasion resistance

1– 6 cm3/50 cm2

Temperature resistance

humid 90°C / dry 140°C

E-glass fiber

Filament diameter

13 µm

Linear density

2400 tex

Change made: The whole manuscript, including tables and figures.

Point 2: Define ASTM in the abstract.

Response 2: It has been defined as follows: American Society for Testing Materials (ASTM).

            NOMENCLATURE

3D FEA

Three-Dimensional finite element analysis

ASTM

the American Society for Testing Materials

Change made: Abstract and nomenclature.

Point 3: Some relevant references in this area are still missing in the introduction section, so include some significant relevant references from recent years. Moderate

Response 3: The following references have been added to the introduction and the list of references:

Vantadori et al. [8] modified Jenq-Shah's two-parameter model (MTPM) [9] to predict the fracture toughness of fiber-reinforced polymers (FRP) using three-point bend specimen according to RILEM [10]. It was found [8] that the predicted values of the fracture toughness and modulus of elastic by MTPM are almost constant; consequently, such parameters are proved to be size-independent. The intrinsic and extrinsic mechanisms of mode I crack growth in FRP have been reviewed by Siddique et al. [11]. They reviewed the main parameters controlling the fracture toughness of such materials. Furthermore, Jia et al. [12] introduced a valuable strategy based on biomimicking to improve the fracture toughness of brittle materials through an intrinsic to extrinsic (ITE) transition. In the ITE transition, toughness started as an intrinsic parameter at the basic material level, but by designing a protein-like effective stress-strain behavior, the toughness at the system level became an extrinsic parameter that increases with the system size with no limit.

---

---

On the other hand, Carpenter et al. [32] confirmed that the matrix under compressive loads follows the relationships established by linear elastic fracture mechanics (LEFM). They found that a log-log plot of the failure load and initial notch length of the experimental data exhibited a linear trend with a slope of -0.54, while the numerical predictions had a linear trend of slope -0.58. As is already known, this plot was expected from LEFM to be linear with a negative slope of 0.5. Liu et al. [33] invoked 3-D finite element analysis (FEA) to study the effect of cohesive zone model (CZM) parameters on the post-buckling and delamination behaviors of FRP under compression. They found that the cohesive strengths mainly affected the unstable delamination stage for the laminates under compression and had little effect on local and global buckling loads. Rozylo [34] concluded that the results obtained based on CZM (numerical) and acoustic emission signals (experimental) showed high agreement. Panettieri et al. [35] used CZM to simulate delaminations growth in compression after impact.

  1. Vantadori, S.; Carpinteri, A.; GÅ‚owacka,K.; Greco, F.; Osiecki, T.; Ronchei, C.; Zanichelli, A. Fracture toughness characterisation of a glass fibre-reinforced plastic composite, Fatigue FractEng Mater Struct. 2021; 44:3–13.
  2. Jenq, Y.; Shah, S. Two-parameter fracture model for concrete. J. Eng. Mech. 1985; 111:1227-1241.
  3. RILEM Technical Committee, 89-FMT. Determination of fracture parameters (KsIC and CTODc) of plain concrete using three-point bend tests, proposed RILEM draft recommendations. Mater. Struct. 1990;23:457-460.
  4. Siddique, A.; Abid, S.; Shafiq, F.; Nawab, Y.; Wang, H.; Shi, B.; Saleemi, S.; Sun, B. Mode I fracture toughness of fiber-reinforced polymer composites: A review, J. Indust.Textiles.2021, Vol. 50(8) 1165–1192.
  5. Jia, Y.; Wang, H.; Liu, B.; Huang,Y.; Gao, H. Intrinsic-to-extrinsic transition in fracture toughness through structural design: A lesson from nature Extreme Mechanics Letters 37 (2020) 100685.

--

--

  1. Carpenter, K.; Lei, Y.; Asadi, A.; Parmigiani, J.; Tabei, A. Applicability of linear elastic fracture mechanics to compressive damage in carbon fiber reinforced epoxy matrix composites, Mech. Adv. Mater. Struct., Published online: 26 Aug 2021. DOI: 10.1080/15376494.2021.1952663.
  2. Liu, P.; Gu, Z.; Peng, X.; Zheng, J. Finite element analysis of the influence of cohesive law parameters on the multiple delamination behaviors of composites under compression, Compos. Struct. 131 (2015) 975–986.
  3. Rozylo, P. Experimental-numerical study into the stability and failure of compressed thin-walled composite profiles using progressive failure analysis and cohesive zone model, Compos. Struct. 257 (2021) 113303.
  4. Panettieri, E.; Fanteria, D.; Danzi, F. Delaminations growth in compression after impact test simulations: Influence of cohesive elements parameters on numerical results, Compos. Struct. 137 (2016) 140–147.

Change made: Introduction and references.

Point 4: Moderate English changes are required in the manuscript.

Response 4: The language of the manuscript has been polished.

Change made: The whole manuscript.

Point 5: The provided Figure's quality is very poor. So, provide the high-resolution Figures.

Response 5: Most figures in the manuscript have been modified according to reviewers' numbers one and two requirements.

Change made: Figures 1 to 4, 6 to 11, and 13, as well as the description in the text.  

Point 6: The text is overlapping in Table 10, arrange them properly.

Response 6: The overlapping in table 10 occurred when the MS word file was converted to pdf format by the journal staff. A double check will be made for the revised version.

Change made: Table 10.

Reviewer 2 Report

The reviewer comments of the paper «An assessment of ASTM E1922 for measuring the Translaminar Fracture Toughness of Laminated Polymer Matrix Composite Materials»

- Reviewer

The authors presented an article «An assessment of ASTM E1922 for measuring the Translaminar Fracture Toughness of Laminated Polymer Matrix Composite Materials». However, there are several points in the article that require further explanation.

Comment 1:

Now the abstract is too focused on detail. The abstract should be written in a more concise way. However, demonstrate in the abstract relevance, practical significance. Add quantitative work results to the abstract.

Comment 2:

On the one hand, the introduction is written logically and clearly. However, the number of articles on the research topic needs to be considered more, especially those on numerical methods for damage evolution and fracture formation of fiber-reinforced composites using Abaqus. Without this, the relevance will not be convincing. Among such articles, it is useful to add an article: doi: 10.1016/j.ceramint.2015.09.085 However, avoid group citations in one phrase.

Comment 3:

Merge Section 2. Description of the Problem into Section 1 Introduction.

Change Section 3. Materials and Methods to Section 2. Materials and Methods, and all the subsections

Comment 4:

Table 1 needs revision to better show the unit and value. 1 – 6 cm3/50 cm2?

Comment 5:

Fig. 2 is very poor, there is no labels, and it cannot be seen clearly.

Comment 6:

Fig. 3 can be improved, add ‘Unit: mm’ in bottom right corner and delete all ‘mm’ shown in the figure, only keep the numbers.

So is Fig. 4.

Comment 7:

The parameters shown in Eqs.1-10 and Table 3 have only been partially defined. All of them need to be defined.

Comment 8:

Illustrate the boundary conditions shown in Fig. 6, both in the text and figure.

Comment 9:

Fig. 9, the legend of contours can hardly be seen clearly, increase the font size.

So is Fig. 10.

Fig. 13, the x and y axis can hardly be seen clearly, increase the font size.

Comment 10:

The English in the article should be significantly improved.

The topic of the article is interesting and deserves attention. It can be seen that the authors have done significant research on the stated topic. However, authors should carefully study all comments. Only after major changes can the article be considered for publication in Polymers

Author Response

Response to Reviewer 2 Comments

General comment:  However, there are several points in the article that require further explanation.

Response: The authors have carefully studied and taken all these points into account to make their article more perspicuously.

Point 1: Now the abstract is too focused on detail. The abstract should be written in a more concise way. However, demonstrate in the abstract relevance, practical significance. Add quantitative work results to the abstract.

Response 1: The abstract has been rewritten according to this suggestion, the first reviewer’s one, and the journal’s limitations; 200 words max, as follows:

The main objective of this work is to predict the exact value of the fracture toughness (KQ) of fiber-reinforced polymer (FRP). The drawback of the American Society for Testing Materials (ASTM) E1922 specimen is the lack of intact fibers behind the crack-tip as in the real case, i.e., through-thickness cracked (TTC) specimen. The novelty of this research is to overcome this deficiency by suggesting unprecedented cracked specimens, i.e., matrix cracked (MC) specimens. This MC exists in the matrix (epoxy) without cutting the glass fibers behind the crack-tip in the unidirectional laminated composite. Two different cracked specimen geometries according to ASTM E1922 and ASTM D3039 were tested. 3-D FEA was adopted to predict the damage failure and geometry correction factor of cracked specimens. The results of the TTC ASTM E1922 specimen showed that the crack initiated perpendicular to the fiber direction up to 1 mm. Failure then occurred due to crack propagation parallel to the fiber direction, i.e., notch insensitivity. As expected, the KQ of the MC ASTM D3039 specimen is higher than that of the TTC ASTM D3039 specimen. The KQ of the MC specimen with two layers is about 1.3 times that of the MC specimen with one layer.

Change made: Abstract.

Point 2:  On the one hand, the introduction is written logically and clearly. However, the number of articles on the research topic needs to be considered more, especially those on numerical methods for damage evolution and fracture formation of fiber-reinforced composites using Abaqus. Without this, the relevance will not be convincing. Among such articles, it is useful to add an article: doi: 10.1016/j.ceramint.2015.09.085 However, avoid group citations in one phrase.

Response 2: unfortunately, the suggested article "DOI: 10.1016/j.ceramint.2015.09.085" is irrelevant. Its title is “Sulfate resistance of ferrochrome slag based geopolymer concrete”, Ceramics International 42(2016) 1254–1260. Besides, several articles on numerical methods for damage evolution and fracture formation of fiber-reinforced composites using ABAQUS have been added to the introduction and the list of references:

They [28] compared the Hashin damage criterion and the eXtended Finite Element Method (XFEM) to predict FRP failure stages. They concluded that the Hashin damage criterion and XFEM predicted the same strength and stiffness of FRP for load levels up to the failure of plies due to matrix cracking.

--

--

On the other hand, Carpenter et al. [32] confirmed that the matrix under compressive loads follows the relationships established by linear elastic fracture mechanics (LEFM). They found that a log-log plot of the failure load and initial notch length of the experimental data exhibited a linear trend with a slope of -0.54, while the numerical predictions had a linear trend of slope -0.58. As is already known, this plot was expected from LEFM to be linear with a negative slope of 0.5. Liu et al. [33] invoked 3-D finite element analysis (FEA) to study the effect of cohesive zone model (CZM) parameters on the post-buckling and delamination behaviors of FRP under compression. They found that the cohesive strengths mainly affected the unstable delamination stage for the laminates under compression and had little effect on local and global buckling loads. Rozylo [34] concluded that the results obtained based on CZM (numerical) and acoustic emission signals (experimental) showed high agreement. Panettieri et al. [35] used CZM to simulate delaminations growth in compression after impact.

  1. Carpenter, K.; Lei, Y.; Asadi, A.; Parmigiani, J.; Tabei, A. Applicability of linear elastic fracture mechanics to compressive damage in carbon fiber reinforced epoxy matrix composites, Mech. Adv. Mater. Struct., Published online: 26 Aug 2021. DOI: 10.1080/15376494.2021.1952663.
  2. Liu, P.; Gu, Z.; Peng, X.; Zheng, J. Finite element analysis of the influence of cohesive law parameters on the multiple delamination behaviors of composites under compression, Compos. Struct. 131 (2015) 975–986.
  3. Rozylo, P. Experimental-numerical study into the stability and failure of compressed thin-walled composite profiles using progressive failure analysis and cohesive zone model, Compos. Struct. 257 (2021) 113303.
  4. Panettieri, E.; Fanteria, D.; Danzi, F. Delaminations growth in compression after impact test simulations: Influence of cohesive elements parameters on numerical results, Compos. Struct. 137 (2016) 140–147.

Change made: Introduction and references.

Point 3: Merge Section 2. Description of the Problem into Section 1 Introduction.
Change Section 3. Materials and Methods to Section 2. Materials and Methods, and all the subsections.

Response 3: The numbers of all sections and subsections have been modified after merging Section 2, “Description of the Problem”, with the introduction. Moreover, Section 2 has been a little more broadly described before being merged  according to point # 2 suggested by reviewer # 3 as follows: 

Panettieri et al. [35] used CZM to simulate delaminations growth in compression after impact.

            The main objective of the present work is to experimentally study the ability of the ASTM E1922 standard test method to measure the real fracture. It is worth noting that two types of cracked ASTM E1922 specimens have been adopted in the present work. The crack types are single edge through-thickness cracked (TTC) specimen (traditional specimen) and single edge matrix cracked (MC) specimen (suggested in the present work as an unprecedented specimen). The pre-crack exists in the matrix without cutting the fibers behind the crack tip in the MC specimen. In the first stage of the experimental work, MC specimens have been used to show the failure at the loading point. While in the second stage, three types of tensile specimens, namely, smooth, TTC, and MC specimens, have been manufactured with dimensions according to ASTM D3039 to measure the exact value of the fracture toughness of GFRE through the MC ASTM D3039 specimen. No such MC specimen in the fibrous composite is suggested before. Therefore, three-dimensional finite element analysis (3D FEA) simulates the composite laminates based on ASTM E 1922 and ASTM D 3039 standard tests. Moreover, Hashin criteria and CIM are used in the present simulation to predict the progressive damage and the geometry correction factor (Y) of the GFRE cracked specimen, respectively.

Change made: Introduction and all sections’ numbers.

Point 4: Table 1 needs revision to better show the unit and value. 1 – 6 cm3/50 cm2?

Response 4: Table 1 has been revised as follows:

Table 1. The physical properties of the used materials.

Material

Physical Properties

Value

Epoxy

Density

1.11 ±0.02 Kg/l

Abrasion resistance

1– 6 cm3/50 cm2

Temperature resistance

humid 90°C / dry 140°C

E-glass fiber

Filament diameter

13 µm

Linear density

2400 tex

Change made: Table 1.

Point 5: Fig. 2 is very poor, there is no labels, and it cannot be seen clearly.

Point 6: Fig. 3 can be improved, add ‘Unit: mm’ in bottom right corner and delete all ‘mm’ shown in the figure, only keep the numbers. So is Fig. 4.

Point 8: Illustrate the boundary conditions shown in Fig. 6, both in the text and figure.

Point 9: Fig. 9, the legend of contours can hardly be seen clearly, increase the font size.
So is Fig. 10. Fig. 13, the x and y axis can hardly be seen clearly, increase the font size.

Responses 5, 6, 8, and 9: Most figures in the manuscript have been modified according to reviewers’ numbers one and two requirements.

Change made: Figures 1 to 4, 6 to 11, and 13, as well as the description in the text.  

Point 7: The parameters shown in Eqs.1-10 and Table 3 have only been partially defined. All of them need to be defined.

Response 7: The parameters shown in Eqs. 1-10 and table 3 have been fully defined as follow:

V is the volume fraction, E is the modulus of elasticity, G is the shear modulus, and ν is the Poisson’s ratio. F is superscript for fiber, and M is superscript for matrix. KF, KM, and K are bulk moduli for fiber, matrix, and composite, respectively, under longitudinal strain. A, B, and C can be obtained from Ref.[37]. Finally, R is a constant.

Table 3 shows the mechanical properties of glass fiber–epoxy lamina, where E11 is the longitudinal modulus of elasticity (parallel to fibers direction). E22 and E33 are the in-plane transverse and out-of-plane moduli of elasticity, respectively. G12 and G13 are the longitudinal shear moduli in two orthogonal planes containing the fibers, and G23 is the out-of-plane shear modulus as well as ν12, ν13, and ν23 are the Poisson’s ratios.

Change made: Section 2.2.1.

Point 10: The English in the article should be significantly improved.

Response 10: The manuscript has been reviewed and reedited to improve grammar, sentence structure, and word spelling.

Change made: The whole manuscript.

General comment:  The topic of the article is interesting and deserves attention. It can be seen that the authors have done significant research on the stated topic. However, authors should carefully study all comments. Only after major changes can the article be considered for publication in Polymers.

Response: The authors would like to thank the reviewer for this comment. Moreover, the authors have carefully studied and taken all comments into account.

Reviewer 3 Report

The work constitutes a current engineering issue. The problems considered in the paper are presented in an interesting way, however, in order for the paper to be published, it is recommended that the authors respond to several of the comments below:

  1. The introduction is described in the correct manner, however, it is recommended that it be strengthened with papers that relate to this research topic: 10.1016/j.compstruct.2015.06.058, 10.1016/j.compstruct.2020.113303 as well as 10.1016/j.compstruct.2015.11.018.
  2. Section 2 should be a little more broadly described.
  3. Figure 1 has very poor quality - please be sure to correct this.
  4. The name of Section 3.2 raises some concerns. Instead of "Numerical work" it would be better to use "Numerical study" or something similar. The same situation applied to name of Section 3.2.3. Instead of "Meshing" it would be better to use "Descrete model".
  5. Please remove the small coordinate systems marked in yellow in Figure 6, because they obscure the figure and are unnecessary since there is a different colored global coordinate system.
  6. The conditions in equations 11-14 are not fully well written, please refer to the paper: Duarte APC, Díaz SA, Silvestre N. Comparative study between XFEM and Hashin damage criterion applied to failure of composites. Thin-Walled Struct. 2017;115:277-288.
  7. Why the results of numerical calculations are presented using the variable SNEG and not ENVELOPE?
  8. Please better describe the qualitative and quantitative evaluation of the study in the conclusion.

Author Response

Response to Reviewer 3 Comments

General Comment: The work constitutes a current engineering issue. The problems considered in the paper are presented in an interesting way, however, in order for the paper to be published, it is recommended that the authors respond to several of the comments below:

Response: The authors have carefully studied and taken all these points into account to improve their article.

Point 1: 
The introduction is described in the correct manner, however, it is recommended that it be strengthened with papers that relate to this research topic: 10.1016/j.compstruct.2015.06.058, 10.1016/j.compstruct.2020.113303 as well as 10.1016/j.compstruct.2015.11.018.

Response 1: The recommended articles are valuable reference papers. These articles and others have been considered in the introduction and added to the list of references as follows:

Vantadori et al. [8] modified Jenq-Shah's two-parameter model (MTPM) [9] to predict the fracture toughness of fiber-reinforced polymers (FRP) using three-point bend specimen according to RILEM [10]. It was found [8] that the predicted values of the fracture toughness and modulus of elastic by MTPM are almost constant; consequently, such parameters are proved to be size-independent. The intrinsic and extrinsic mechanisms of mode I crack growth in FRP have been reviewed by Siddique et al. [11]. They reviewed the main parameters controlling the fracture toughness of such materials. Furthermore, Jia et al. [12] introduced a valuable strategy based on biomimicking to improve the fracture toughness of brittle materials through an intrinsic to extrinsic (ITE) transition. In the ITE transition, toughness started as an intrinsic parameter at the basic material level, but by designing a protein-like effective stress-strain behavior, the toughness at the system level became an extrinsic parameter that increases with the system size with no limit.

---

On the other hand, Carpenter et al. [32] confirmed that the matrix under compressive loads follows the relationships established by linear elastic fracture mechanics (LEFM). They found that a log-log plot of the failure load and initial notch length of the experimental data exhibited a linear trend with a slope of -0.54, while the numerical predictions had a linear trend of slope -0.58. As is already known, this plot was expected from LEFM to be linear with a negative slope of 0.5. Liu et al. [33] invoked 3-D finite element analysis (FEA) to study the effect of cohesive zone model (CZM) parameters on the post-buckling and delamination behaviors of FRP under compression. They found that the cohesive strengths mainly affected the unstable delamination stage for the laminates under compression and had little effect on local and global buckling loads. Rozylo [34] concluded that the results obtained based on CZM (numerical) and acoustic emission signals (experimental) showed high agreement. Panettieri et al. [35] used CZM to simulate delaminations growth in compression after impact.

  1. Vantadori, S.; Carpinteri, A.; GÅ‚owacka,K.; Greco, F.; Osiecki, T.; Ronchei, C.; Zanichelli, A. Fracture toughness characterisation of a glass fibre-reinforced plastic composite, Fatigue FractEng Mater Struct. 2021; 44:3–13.
  2. Jenq, Y.; Shah, S. Two-parameter fracture model for concrete. J. Eng. Mech. 1985; 111:1227-1241.
  3. RILEM Technical Committee, 89-FMT. Determination of fracture parameters (KsIC and CTODc) of plain concrete using three-point bend tests, proposed RILEM draft recommendations. Mater. Struct. 1990;23:457-460.
  4. Siddique, A.; Abid, S.; Shafiq, F.; Nawab, Y.; Wang, H.; Shi, B.; Saleemi, S.; Sun, B. Mode I fracture toughness of fiber-reinforced polymer composites: A review, J. Indust.Textiles.2021, Vol. 50(8) 1165–1192.
  5. Jia, Y.; Wang, H.; Liu, B.; Huang,Y.; Gao, H. Intrinsic-to-extrinsic transition in fracture toughness through structural design: A lesson from nature Extreme Mechanics Letters 37 (2020) 100685.

--

--

  1. Carpenter, K.; Lei, Y.; Asadi, A.; Parmigiani, J.; Tabei, A. Applicability of linear elastic fracture mechanics to compressive damage in carbon fiber reinforced epoxy matrix composites, Mech. Adv. Mater. Struct., Published online: 26 Aug 2021. DOI: 10.1080/15376494.2021.1952663.
  2. Liu, P.; Gu, Z.; Peng, X.; Zheng, J. Finite element analysis of the influence of cohesive law parameters on the multiple delamination behaviors of composites under compression, Compos. Struct. 131 (2015) 975–986.
  3. Rozylo, P. Experimental-numerical study into the stability and failure of compressed thin-walled composite profiles using progressive failure analysis and cohesive zone model, Compos. Struct. 257 (2021) 113303.
  4. Panettieri, E.; Fanteria, D.; Danzi, F. Delaminations growth in compression after impact test simulations: Influence of cohesive elements parameters on numerical results, Compos. Struct. 137 (2016) 140–147.

Change made: Introduction and references.

Point 2: Section 2 should be a little more broadly described.

Response 2: Section 2 has been a little more broadly described. According to point # 3 suggested by reviewer # 2, "Merge Section 2 into Section 1", sections 1 and 2 have been merged as follows: 

Panettieri et al. [35] used CZM to simulate delaminations growth in compression after impact.

            The main objective of the present work is to experimentally study the ability of the ASTM E1922 standard test method to measure the real fracture. It is worth noting that two types of cracked ASTM E1922 specimens have been adopted in the present work. The crack types are single edge through-thickness cracked (TTC) specimen (traditional specimen) and single edge matrix cracked (MC) specimen (suggested in the present work as an unprecedented specimen). The pre-crack exists in the matrix without cutting the fibers behind the crack tip in the MC specimen. In the first stage of the experimental work, MC specimens have been used to show the failure at the loading point. While in the second stage, three types of tensile specimens, namely, smooth, TTC, and MC specimens, have been manufactured with dimensions according to ASTM D3039 to measure the exact value of the fracture toughness of GFRE through the MC ASTM D3039 specimen. No such MC specimen in the fibrous composite is suggested before. Therefore, three-dimensional finite element analysis (3D FEA) simulates the composite laminates based on ASTM E 1922 and ASTM D 3039 standard tests. Moreover, Hashin criteria and CIM are used in the present simulation to predict the progressive damage and the geometry correction factor (Y) of the GFRE cracked specimen, respectively.

Change made: Introduction and all sections' numbers.

Point 3: Figure 1 has very poor quality - please be sure to correct this.

Response 3: Figure 1 has been corrected and modified.

Change made: Figure 1.

Point 4: The name of Section 3.2 raises some concerns. Instead of "Numerical work" it would be better to use "Numerical study" or something similar. The same situation applied to name of Section 3.2.3. Instead of "Meshing" it would be better to use "Descrete model".

Response 4: They have been changes according to the reviewer's suggestions:

Change made: Numerical study.

Point 5: Please remove the small coordinate systems marked in yellow in Figure 6, because they obscure the figure and are unnecessary since there is a different colored global coordinate system.

Response 5: Considering the reviewer's suggestion, Fig. 6 has been modified.

Change made: Figure 6

Point 6: The conditions in equations 11-14 are not fully well written, please refer to the paper: Duarte APC, Díaz SA, Silvestre N. Comparative study between XFEM and Hashin damage criterion applied to failure of composites. Thin-Walled Struct. 2017; 115:277-288.

Response 6: The recommended article has been used to explain Hashin's damage equations in more detail.

Change made: Section 2.2.4 Hashin Damage Model

Point 7: Why the results of numerical calculations are presented using the variable SNEG and not ENVELOPE?

Response 7:  The authors would like to thank the reviewer for this comment. Since the SNEG variable is the ABAQUS's default, the authors have used it and have not tried ENVELOPE before. Once again, the authors thank the reviewer for getting their attention to present the numerical results using ENVELOPE.

Change made: ----

Point 8: Please better describe the qualitative and quantitative evaluation of the study in the conclusion.

Response 8: The conclusions have been modified as follows:

Change made: Conclusions

Round 2

Reviewer 2 Report

Thank you for the revisions. However, I was hoping that the authors can pay more attention to the previous comments, so that this second revision can be avoided. I have always been very careful to avoid low-level mistakes. The acuthors have referred to the wrong article, even if they have realised, but they didn't double check the original comments. Obviously 'Sulfate resistance of ferrochrome slag based geopolymer concrete' is irrelevant because its doi is ..058 not ..085 as pointed out in the original comments. 

Author Response

Response to Reviewer 2 Comment

General comment:  Thank you for the revisions. However, I was hoping that the authors can pay more attention to the previous comments, so that this second revision can be avoided. I have always been very careful to avoid low-level mistakes. The authors have referred to the wrong article, even if they have realised, but they didn't double check the original comments. Obviously 'Sulfate resistance of ferrochrome slag based geopolymer concrete' is irrelevant because its doi is ..058 not ..085 as pointed out in the original comments.

Response: The authors would like to apologize for this inadvertent error, and the correct article "Design and analysis of the porous ZrO2/(ZrO2+Ni) ceramic joint with load bearing–heat insulation integration" has been added to the introduction as follows:

Zhou et al. [36] used the shear damage initiation criterion, available in ABAQUS/explicit, to model the shear failure due to fracture within shear bands in metal-ceramic functionally graded bolted joint. It is worth noting that the bolt was made of porous ZrO2/(ZrO2+Ni) FGMs. They used Tsai–Wu tensor theory as the failure criteria of the C/SiC plates. They concluded that ZrO2+15 vol% Ni of two mm thickness is the optimal shear band to balance such bolted joint's shearing strength and heat insulation performance.

  1. Zhou, W.; Zhang, R.; Fang, D. Design and analysis of the porous ZrO2/(ZrO2+ Ni) ceramic joint with load bearing–heat insulation integration. Ceramics International 2016, 42, 1416-1424.

Change made: Introduction and references.

Round 3

Reviewer 2 Report

Thank you for the response, I have no further comments.